# Research Progress on NSP11 of Porcine Reproductive and Respiratory Syndrome Virus

**DOI:** 10.3390/vetsci10070451

**Published:** 2023-07-10

**Authors:** Yajie Zheng, Hang Zhang, Qin Luo, Huiyang Sha, Gan Li, Xuanru Mu, Yingxin He, Weili Kong, Anfeng Wu, Haoji Zhang, Xingang Yu

**Affiliations:** 1School of Life Science and Engineering, Foshan University, Foshan 528231, China; zhengyajie2022@163.com (Y.Z.); hangzh2022@163.com (H.Z.); luoqin121104@163.com (Q.L.); huiyangsha2022@163.com (H.S.); ligan1227@163.com (G.L.); 15940815092@163.com (X.M.); hyxin@outlook.com (Y.H.); zhanghaoji@aliyun.com (H.Z.); 2Gladstone Institutes of Virology and Immunology, University of California, San Francisco, CA 94158, USA; weili.kong@gladstone.ucsf.edu; 3Maccura Biotechnology Co., Ltd., Chengdu 510000, China; 18813291567@163.com

**Keywords:** porcine reproductive and respiratory syndrome virus, NSP11, genetic evolution, viral replication, protein interactions, enzyme activity

## Abstract

**Simple Summary:**

NSP11 is a non-structural protein of the porcine reproductive and respiratory syndrome virus (PRRSV) that influences viral replication and interacts with the host’s innate immune response. We reviewed the genetic evolution of NSP11, its effects on PRRSV replication and virulence, its interactions with other PRRSV and host proteins, the regulation of host immunity, the conserved characteristics of nidovirus-specific endonuclease (NendoU), and its diagnosis, providing an important theoretical basis for in-depth studies on PRRSV pathogenesis and vaccine design.

**Abstract:**

Porcine reproductive and respiratory syndrome (PRRS) is a virulent infectious disease caused by the PRRS virus (PRRSV). The non-structural protein 11 (NSP11) of PRRSV is a nidovirus-specific endonuclease (NendoU), which displays uridine specificity and catalytic functions conserved throughout the entire NendoU family and exerts a wide range of biological effects. This review discusses the genetic evolution of NSP11, its effects on PRRSV replication and virulence, its interaction with other PRRSV and host proteins, its regulation of host immunity, the conserved characteristics of its enzyme activity (NendoU), and its diagnosis, providing an essential theoretical basis for in-depth studies of PRRSV pathogenesis and vaccine design.

## 1. Introduction

Porcine reproductive and respiratory syndrome (PRRS) is a prevalent and highly contagious disease caused by the porcine reproductive and respiratory syndrome virus (PRRSV). Pregnant sows are at a higher risk of experiencing negative reproductive outcomes such as miscarriage, premature birth, and stillbirth. Additionally, respiratory symptoms are commonly observed in fattening pigs and piglets [1,2]. Globally, PRRSV infections are at an epidemic level, underlying substantial economic losses in the pig industry.

PRRSV belongs to the *Arteriviridae* family, which includes equine arteritis virus (EAV), lactate dehydrogenase-elevating virus (LDV), and simian hemorrhagic fever virus (SHFV) [3,4,5]. PRRSV is divided into two genotypes, the European or type 1 virus and the American or type 2 virus [6]. The PRRSV genome encodes at least 10 open reading frames (ORFs) [7]. ORF1a and ORF1b occupy the 5′ proximal 75% of the genome and encode two polyproteins—pp1a and pp1ab. Viral-encoded proteases are responsible for processing the pp1a and pp1ab polyproteins, which subsequently generate 16 non-structural proteins (NSPs) (Figure 1). These proteins are crucial for the replication and transcription of the viral RNA genome [8].

PRRS was initially identified in America as a new pig disease that can cause delayed reproductive failure and severe pneumonia in newborn piglets [9]. The disease was first identified in the Midwest region of America in 1987, after which it spread to Canada, France, Japan, and other countries [10]. In 1996, PRRSV was first isolated from breeding pigs in China. Subsequently, pig farms situated in northern and coastal regions of China have encountered noticeable rates of infection [11,12]. Since 2006, the highly pathogenic PRRSV (HP-PRRSV) has been extensively disseminated throughout China, resulting in significant economic losses within the pig farming industry [13,14]. PRRSV has exhibited rapid evolution, and the latest variant of PRRSV-2 in China shows higher virulence than all previous strains [15,16]. Xie et al. [17] conducted an epidemiological survey of 2428 pig farms across 27 provinces in China. The rates of positivity of PRRSV varied among the different regions, ranging from approximately 8.12% to 29.33%. Guo et al. [18] systematically analyzed the epidemic status of PRRSV in China from a molecular epidemiology perspective. At present, PRRSV-1 and PRRSV-2 are spreading, and about 80% of pig farms are seropositive for PRRSV. PRRSV has spread globally, and the current vaccines for preventing and controlling PRRSV have not proven sufficiently effective [19]. In addition to vaccination, implementing effective biosecurity measures has been deemed crucial for PRRS control in European countries and elsewhere. A study in Italy demonstrated that coupling immunity with biosecurity is highly effective at curbing viral transmission and restricting the virus’ prevalence within pig populations [20]. Studies in Saxony and Thuringia have also validated that the employment of biosecurity measures to regulate animal (segregating contaminated and non-contaminated areas) and personnel (recording visitors) movements helps to mitigate PRRSV infection [21].

NSP11 is expressed as part of the large polyprotein pp1ab, which is directly translated from the viral genome pp1ab (Figure 1). The protein consists of 223 amino acids and features a distinctive NendoU domain, which is notably conserved among viruses in the *Nidovirales* order. [22]. NSP11 is a key genetic marker of the Nidovirales virus, which can infect vertebrate hosts. This marker is encoded by ORF1b [23,24]. PRRSV NSP11 exhibits uridylate-specific RNA cleavage and consists of subdomain A and subdomain B. Subdomain A harbors the nuclease activity, whereas subdomain B determines the overall structural conformation. A newly determined crystal structure revealed that NSP11 assembles into an asymmetric dimer [25], differing from the hexametric structure of coronavirus NSP15 [26,27]. The NendoU domain resides in the C-terminal region of NSP11 and is distantly related to the XendoU family. XendoU is an endoribonuclease derived from *Xenopus laevis* [5,28,29]. The NendoU homolog XendoU has been reported to be a monomer [30]. Studies have shown that NendoUs, including PRRSV NSP11, have the same ancestor, XendoU. XendoU and NendoU of PRRSV NSP11 were found to have similar structural features in their C-terminal domains (CTDs) but they display some differences in their catalytic mechanisms [31]. Comparing PRRSV NSP11’s NendoU with XendoU can provide a deeper understanding of the endoribonuclease family.

NSP11 is a key player in PRRSV biology and a potential target for prevention measures [32]. This review focuses on the genetic evolution of NSP11, its effects on PRRSV replication and virulence, its interactions with other PRRSVs and host proteins, the regulation of host immunity, the conserved characteristics of enzyme activity (NendoU), and its diagnosis, providing an important theoretical basis for in-depth studies of PRRSV pathogenesis and vaccine design. These will also provide a reference for clinical PRRSV prophylactic strategies.

## 2. Genetic Evolution Analysis of the NSP11 Sequence

To analyze the NSP11 sequence conservation, 36 PRRSV NSP11 sequences were selected from the NCBI nucleotide database (Table 1). The strains selected included strains identified in various years spanning 1996 through 2022, vaccine strains, and commonly cited representative strains.

To determine the similarities among NSP11 nucleotide sequences from different PRRSV lineages, we analyzed the NSP11 nucleotide sequences of reference strains using Clustal W in the MegAlign function with DNAStar software (version 7.0). The nucleotide homology rates between PRRSV-1 and PRRSV-2 ranged from 64.3% to 100%. The strains with a nucleotide homology rate of 64.3% included 10FS-GD-2010/China/2010 and BJEU06-1/China/2006, as well as HeNan-A9/China/2013 and BJEU06-1/China/2006. All PRRSV-2 strains, including 10FS-GD-2010/China/2010, HeNan-A9/China/2013, GD-2007/Australia/2007, and HH08/China/2011, displayed 100% nucleotide homology. The nucleotide homology rates among the PRRSV-1 strains ranged from 80.4% to 92.8%, with strains including ES13-49-P85/Spain/2013 and WestSib13/Russia/2013 displaying 80.4% nucleotide homology and strains including BJEU06-1/China/2006 and LV4.2.1/Netherlands/2004 displaying 92.8% nucleotide homology. The nucleotide homology rates among the PRRSV-2 strains ranged from 84.8% to 100%, with strains including CH-HNPY-01-2022/China/2022 and MN184A/USA/2001 displaying 84.8% nucleotide homology (Figure 2). These nucleotide homology analyses suggest that PRRSV-1 and PRRSV-2 share low intergroup nucleotide homology but display high intragroup nucleotide homology. These results reflect significant differences in NSP11 between PRRSV-1 and PRRSV-2, with conserved NSP11 sequences within each group.

According to the phylogenetic analysis based on the nucleotide sequence of PRRSV NSP11, LV4.2.1/Netherlands/2004 and BJE06-1/China/2006 strains of PRRSV-1 are located in the same branch with a short genetic distance. CH-1a/China/1996 and Ingelvac ATP/USA/1999 (vaccine strains) belong to PRRSV-2 strains within the phylogenetic tree. JXA1/China/2006, a highly pathogenic PRRSV-2 strain, was genetically close to JX143/China/2006 and HUB1/China/2006. Moreover, some PRRSV-2 strains from different countries, including A2MC2-2017/USA/2017 and CC-1/China/2006, were genetically closely related, providing indirect evidence for the transmission of PRRSV between countries around the world (Figure 3).

The nucleotide sequence alignment and phylogenetic tree analyses indicated that PRRSV NSP11 is relatively conserved compared to the highly variable PRRSV NSP2 [33]. Chen et al. [34] passaged the HP-PRRSV XH-GD strain in Marc-145 cells 122 consecutive times. Thirty-five amino acid changes in structural and non-structural proteins were observed within these 122 passages, and the viral virulence weakened during passaging. A mutation occurred in NSP11 at amino acid position 1238 (amino acid position numbering is based on the sequence of the XH-GD strain). These studies indicate that as the passage number increases, the viral virulence decreases, implying that mutations in NSP11 amino acid sites may be involved in the modulation of viral virulence during passage.

## 3. NSP11 Participates in Viral Replication

In the complex pathogenic mechanism of PRRSV, NSPs play a critical role in regulating virus replication and pathogenicity and antagonizing host antiviral immune regulation [35,36,37,38]. PRRSV RNA NendoU NSP11 belongs to the NendoU superfamily and contributes to *Arterivirus* replication [25].

In their study, Wen et al. [39] investigated the impact of p21 knockdown through RNA interference on cell cycle progression using cytometry. They found that NSP11 plays a role in PRRSV replication through its RNA processing function. The study indicates that PRRSV infection leads to the degradation of p21 and prompts Marc-145 cells to enter the S phase of the cell cycle. The p21 degradation occurs through a proteasome-dependent mechanism that is not reliant on ubiquitination but instead requires the NendoU activity of NSP11. NSP11 has been shown to hinder cell proliferation, ultimately enhancing the virus’ ability to utilize the cell’s nutrients for its propagation. An additional experiment provided evidence that the RNA processing function of NSP11 correlates directly with viral replication. Shi et al. [37] created a NSP11 expression plasmid and introduced it into cells, demonstrating that NSP11 overexpression led to increased PRRSV titers in Marc-145 cells, whereas the transfection of a NSP11-specific siRNA decreased PRRSV titers in Marc-145 cells. In a subsequent study, Sun et al. [40] found that NSP11 caused a delay in cell cycle progression during the S phase, as evidenced by flow cytometry. Additionally, NSP11 overexpression resulted in cell cycle arrest, as confirmed by BrdU staining. NSP11’s NendoU activity was found to be responsible for processing RNA, which ultimately affects viral replication. This finding suggests that targeting NSP11 may be a viable strategy for controlling PRRSV replication.

Wen et al., in a study of the NSP11 function using P21, transfection of NSP11-specific siRNA, and overexpression of NSP11, confirmed that PRRSV NSP11 participates in and affects viral replication. PRRSV NSP11 can participate in and affect viral replication. In addition, the special NendoU structure of PRRSV NSP11 directly determines the important role NSP11 plays in normal PRSSV life activities. In-depth studies of PRRSV NSP11 are necessary for the future prevention and control of PRRS and for the research and development of related drugs and vaccines. Therefore, we speculate that inhibitors targeting NSP11 may become powerful tools for addressing PRRSV infection.

## 4. Virulence Effect of NSP11

Kwon et al. [41] conducted a study using a sow reproductive failure model and found that unstructured genomic regions (ORF1a and 1b) and structured genomic regions (ORF2-7) may impact PRRSV virulence. Their study concluded that NSP3-8 and ORF5 are the primary locations of major virulence determinants, while other determinants may also be present in NSP1-3, NSP10-12, and ORF2. NSP11 is a potential virulence determinant of PRRSV.

Li et al. [35] conducted additional virulence studies by creating full-length infectious cDNA clones with swapped coding regions between the highly pathogenic RvJXwn and minimally pathogenic RvHB-1/3.9. They substituted the NSP9-, NSP10-, NSP11-, and NSP12-coding regions separately; NSP9- and NSP10-coding regions together; or NSP9-, NSP10-, and NSP11-coding regions simultaneously between the two viruses. This study revealed that all four single-region substitutions of the rescued virus exhibited significantly higher viral titers at multiple time points compared to the parental virus RvHB-1/3.9. However, RvHJn9n10 showed slightly lower viral titers than RvHB-1/3.9. These results show that the HP-PRRSV NSP9 and NSP10 coding regions are closely associated with the replication efficiency in vitro and in vivo, and with increased pathogenicity and lethality in piglets. This study found that substitutions in individual regions resulted in higher viral titers than the parent virus, suggesting that the substitutions in NSP11 did not diminish the virus’ virulence. Another experiment conducted by Jiang et al. [42] involved replacing the non-structural protein coding regions NSP9, NSP10, and NSP11 separately. Replacing NSP9, NSP10, and NSP11 alone did not affect the virus’ virulence.

The current research indicates that replacing NSP11 alone or simultaneously replacing NSP11 and NSP10 does not affect PRRSV virulence. However, the study by Li et al. mentioned above found that replacing NSP9 or NSP10 alone did not affect the viral pathogenicity but replacing NSP9 and NSP10 at the same time did affect the viral virulence. We, thus, speculate that other factors, including structural proteins or non-structural PRRSV proteins, may jointly affect viral virulence with NSP11. Extensive research is required to confirm this.

## 5. NSP11 Interacts with PRRSV Proteins

Song et al. [43] identified interactions between PRRSV NSPs using yeast two-hybrid (Y2H) screening combined with coimmunoprecipitation (CO-IP) experiments. They identified seven pairs of non-structural proteins that interacted bidirectionally, including NSP9-NSP11 and NSP11-NSP12. According to the Y2H screen conducted in this study, NSP11 was found to be the second most interactive non-structural protein. Additionally, this study revealed that NSP11 can interact with other PRRSV NSPs, including NSP12, NSP2, NSP5, NSP6, and NSP9, indicating strong connections between NSP11 and other PRRSV NSPs. This study also suggests that NSP9, NSP10, NSP11, and NSP12 may be key components of the PRRSV replication–transcription complex (RTC). NSP11 and NSP12 display a significant interaction, suggesting that NSP11 may be recruited to the RTC through NSP12 to perform unknown biological functions.

In the study conducted by Song et al., the Y2H system failed to detect any multimerization of NSP11. In contrast, Shi et al. found that NSP11 can effectively perform related biological functions upon multimerization [25]. In a subsequent experiment, Nan et al. [44] provided further evidence of the interaction between NSP11 and NSP12 and of NSP11 multimerization.

The above studies show that NSP11 can interact with NSP2, NSP5, NSP6, NSP9, and NSP12, with a particularly strong interaction observed between NSP11 and NSP12. This interaction promotes the recruitment of NSP11 to the RTC and elicits certain biological functions. Understanding the interaction mechanism between NSP11 and other NSPs is crucial to the development of drugs to target NSP11. PRRSV replication may be inhibitable by manipulating NSP11 and disrupting the normal biological functions of other NSPs.

## 6. NSP11 Interacts with Host Proteins

Viral proteins create a favorable environment for viral replication by interacting with host cell proteins, causing severe diseases in the host. Identifying important interactions between host proteins and viral proteins is a key step in investigating viral protein functions and their roles in replication. Interactions between host cell proteins and PRRSV NSP11 are important for viral replication and pathogenicity, and PRRSV NSP11 influences protein metabolism, cell signal transduction, and cell pathogenicity [45].

Several host cellular proteins exert antiviral activity. PRRSV NSP11 inhibits or degrades cellular proteins including interleukin-1 receptor-associated kinase 1 (IRAK1), proprotein convertase subtilisin/kexin-9 (PCSK9), tripartite motif-containing 59 (TRIM59), signal transducer and activator of transcription 2 (STAT2), and cholesterol-25-hydroxylase (CH25H), hindering their biological functions. NSP11 inhibits IFN-β production by inhibiting IRAK1 expression. Amino acid residue K59 in NSP11 is crucial to its ability to downregulate STAT2. Thus, NSP11 can antagonize IFN signaling by mediating STAT2 degradation, providing a deeper understanding of how PRRSV interferes with innate immunity mechanisms [46,47]. The antiviral function of CH25H is also affected by PRRSV NSP11. A study by Dong [48] et al. found that NSP11 mediates CH25H degradation through the lysosomal pathway. NSP11 antagonizes CH25H’s anti-PRRSV activity by degrading CH25H. In addition, NSP11 has NendoU activity. This activity enables NSP11 to dose-dependently inhibit PCSK9 expression. PCSK9 can inhibit PRRSV replication, while NSP11 indirectly promotes PRRSV replication by inhibiting PCSK9 [32,49]. TRIM59, which has antiviral activity, is also inhibited by PRRSV NSP11. Jing et al. [50] observed a potential interaction between TRIM59 and NSP11. Studies have confirmed that the N-terminal loop structure of TRIM59 interacts with the C-terminal NendoU domain of NSP11, and NSP11 can counteract TRIM59’s antiviral function.

PRRSV NSP11 promotes viral replication by inducing the overexpression of heat shock protein 27 (HSP27) and mucosa-associated lymphoid tissue lymphoma translocator 1 (MALT1). HSP27 plays a multifunctional role in viral replication as a cellular chaperone. A CO-IP analysis conducted by Song et al. [51] screened for interactions between HSP27 and PRRSV-NSPs. HSP27 was noted to interact with various PRRSV NSPs, including NSP11, which can promote PRRSV replication by interacting with HSP27. Han et al. [52] found that NSP11 can induce MALT1 synergistically with NSP7β and NSP4 and promote PRRSV replication through MALT1 proteolytic activity. In contrast to the promotion of viral replication caused by PRRSV NSP11-induced cell protein, PRRSV NSP11-induced galactose lectin-1 (Gal-1) can inhibit PRRSV replication. Gal-1 overexpression has been shown to inhibit the replication of multiple PRRSV strains, whereas Gal-1 knockdown or knockout has been shown to increase viral titer and nucleocapsid protein expression. A study conducted by Li et al. [53] showed that Gal-1 interacts with the NendoU domain of NSP11 to induce Gal-1 and thereby inhibit PRRSV proliferation.

NSP11 has been shown to interact with interferon regulators, inhibiting the transmission of interferon signals. Its interaction with interferon regulatory factor 9 (IRF-9) inhibits the interferon-stimulated nuclear transfer of gene factor 3, ultimately blocking the transduction of IFN-I signals [54]. A study conducted by Dong [48] found that NSP11 can also downregulate IRF-1 expression. His-129, His-144, and Lys-173 of NSP11 are key amino acids for downregulating IRF-1. IRF-1, a cell regulator, has been shown to inhibit PRRSV replication [55,56]. However, IRF-1 downregulation by NSP11 can inhibit various immune processes [57]. OTU deubiquitinase with linear linkage specificity (OTULIN) assists NSP11 in enhancing its biological function and increasing its participation in host innate immunity. A study conducted by Su et al. [58] discovered that the OTU deubiquitinase with OTULIN and synergistic effect can enhance the biological functions of NSP11. They observed that PRRSV NSP11 was able to recruit OTULIN through non-enzymatic binding, which consequently increased its efficacy in eliminating the linear ubiquitination of nuclear factor kappa-B (NF-κB)-essential modulators and inhibiting IFN-I production. The synergistic effect of porcine OTULIN and PRRSV NSP11 reduces the level of cellular protein ubiquitin, which is related to innate immunity.

PRRSV NSP11 interacts with multiple host proteins, including IRAK1, STAT2, OTULIN, IRF9, and IRF-1, as depicted in Figure 4. This interaction enables NSP11 to act as a NendoU and modulate the host’s innate immune response, ultimately impacting PRRSV replication.

## 7. NSP11 Participates in Host Innate Immune System

The PRRSV NSP11 research primarily focuses on its effects on immune processes within host cells. PRRSV is an immunosuppressive pathogen inducing poor innate immune responses, especially by suppressing type I interferons (IFNs) [59,60]. Pigs infected with PRRSV show low levels of IFN-α, including undetectable IFN-α in the lungs [61,62]. Previous studies verified that several PRRSV NSPs, including NSP1α and-1β [63], NSP2 [64], NSP4 [65], and NSP11 [22,47], induce host immunosuppression by inhibiting IFN. NSP11 of PRRSV exhibits nidovirus-like NendoU activity, which is very important for the viral replication and inhibition of a host’s innate immune system [50]. PRRSV NSP11 has been found to play a role in IFN inhibition [66].

The NSP11 protein significantly inhibits the activity of the IFN-β promoter and IRF3 reporter gene. This inhibition was caused in the presence of dsRNA, and IRF3 can also be triggered as an inducer. Furthermore, PRRSV NSP11 blocks IRF3 (an antiviral transcription factor) activation. By inhibiting IRF3, NSP11 evades the host’s innate immune system [67]. Beura et al. [68] confirmed that NSP11 inhibits the activation of the IFN-β promoter, and Montaner-Tarbes et al.’s [69] research further supports this finding. Wang et al. [70] confirmed that NSP11 and NSP1 collectively antagonize the production of IFN-I, particularly IFN-β, with similar functions. In addition, the study found that the activity of NSP11’s deubiquitinating enzyme (DUB) weakens the host’s innate immune response by inhibiting NF-κB activation. PRRSV NSP11 can also affect the host’s innate immune system by affecting interleukin-10 (IL-10) and IL-17. Burgara-Estrella et al. [71] utilized bioinformatics prediction methods to scan amino acid sequences and identify potential T cell epitopes. NSP11 plays a role in the innate immune system of the host by impacting IL-10. In their study, Wang et al. [72] discovered that PRRSV exerts a significant impact on IL-17 expression. They also identified Ser74 and Phe76 in NSP11 as crucial epitopes for IL-17 production and viral replication. This study generated a mutant lacking aa 47 to 92 and transfected it into 3D4/21 cells. NSP11 lacking aa 47 to 92 did not upregulate IL-17 expression. This indicates that the region of aa 47 to 92 in NSP11 is crucial for IL-17 production. This study further constructed NSP11 mutants in which serine (S74A), phenylalanine (F76A), or both were replaced by alanine. S74A, F76A, and S74A/F76A mutants failed to induce IL-17 expression (and activate the IL-17 promoter), indicating that these two residues are essential for NSP11 to induce IL-17 production.

PRRSV NSP11 induces host innate immune responses. Jiang et al. [73] utilized supernatants from immunized mice or hybridoma cultures to identify antibodies against the NSP11 protein in serum. A specific region on the surface of NSP11 was identified as the B-cell epitope. 111DCREY115 was found to be the core unit of the B-cell epitope recognized by mAb 3F9. When targeted by specific monoclonal antibodies, these B-cell epitopes of NSP11 are able to induce humoral immune responses in pigs infected with PRRSV. These results suggest that NSP11 plays a role in the host’s innate immune response and can trigger a humoral immune response in the host. This study analyzed the PRRSV proteins responsible for the differential expression rates of TNF-α in PAMs. NSP7, NSP11, and NSP12 were found to be crucial for inducing differential TNF-α mRNA levels in various PRRSV strains. Additionally, this study found that NSP1β and NSP11 of PRRSV could inhibit the ERK signaling pathway, ultimately inhibiting TNF-α production. The roles of NSP1β and NSP11 in differential TNF-α mRNA expression and their contributions to the pathogenesis of PRRSV have been confirmed through reverse genetics [74,75].

The research described above provides ample evidence that PRRSV NSP11 can participate in the host’s innate immune response. The relationship between PRRSV NSP11 and host innate immunity is undoubtedly a focus of research with respect to PRRSV NSP11 function. Through the above research on interactions between PRRSV NSP11 and various cytokines participating in and affecting host innate immunity, we found that NSP11 can participate in various cellular pathways to influence the normal transmission of biological information in cells. For example, interacting with IRF3 helps PRRSV escape the host’s innate immune system, inhibiting ERK signaling, resulting in TNF-α suppression, impacting IL-17, and indirectly impacting the host’s natural immune system. The immune escape of PRRSV is an important reason why PRRS is difficult to prevent and control. NSP11 as a drug design target may be a solution to PRRSV immune evasion.

## 8. Conservative Characteristics of NSP11 Enzyme Activity (NendoU)

As an *Arterivirus* component, PRRSV NSP11 has a unique structure (dimer) and conservative biochemical function. It encodes a unique and conservative NendoU in the *Nestoridae* family. Subdomain A of NSP11 contains an active site for endoribonuclease activity, whereas subdomain B maintains the overall structure of NSP11. It is also a component of the RNA-dependent RNA polymerase (RdRp) complex used in viral RNA synthesis [8,76]. The NendoU activity of NSP11 is conserved and unique to the *Nidovirales* sequence [5]. Furthermore, PRRSV NendoU in NSP11 has uridylate-specific RNA cleavage activity. Unlike coronavirus NSP15, which forms a hexagonal structure to achieve the greatest enzyme activity [77], PRRSV NSP11 exists as a homodimer that stimulates nuclease activity [25]. The NendoU activity of NSP11 allows it to antagonize the antiviral activity of proprotein convertase subtilisin/kexin type 9 (PCSK9) [78] and inhibit IFN-β induction.

The NendoU domain of PRRSV NSP11 is conserved. Hong et al. [79] found that NSP11 specifically removes the polyubiquitin chain linked with lysine 48 (K48), and its conserved sites C112, H144, D173, K180, and Y219 are essential for its DUB activity. Its biochemical characterization revealed the conservation of NendoU in the range of nested viruses. The biochemical and structural characterization of NendoU may provide important clues for developing specific anti-PRRSV drugs. Nedialkova et al. [78] investigated the impact of replacing conservative amino acids in the EAV tender bean domain on RNA cleavage and substrate specificity. This study found that the double substitution of His-126 and His-141 or substitutions of Lys-170 or Tyr-216 with Ala resulted in undetectable levels of RNA2 cleavage. These findings support the crucial role of mutant residues in *Arterivirus*-mediated RNA processing. The NendoU region is conserved to protect viral RNA from degradation.

The representative strain CH-1a was selected as a reference strain and used to model a three-dimensional protein structure of NSP11 based on previous studies of the structure of PRRSV NSP11 (Figure 5). The amino acid sequence of PRRSV NSP11 was obtained from NCBI and imported into the SWISS-MODEL online software for three-dimensional structure predictions. The reliability and quality of the predicted results were evaluated using the GMQE and QMEANDisCo Global algorithms. The GMQE score was 0.92 and the QMEANDisCo Global score was 0.84 ± 0.05, indicating high accuracy and reliability of the predicted protein structure. The predicted results showed a sequence identity of 97.76% with the PRRSV NSP11 protein predicted by Shi et al. [25]. These results demonstrate the high reliability of the current prediction. The predicted three-dimensional structure shows the N-terminal and C-terminal domains of PRRSV NSP11. The NendoU motif of PRRSV NSP11 is located in the C-terminal domain (Figure 5).

The above research on the NendoU structure within PRRSV NSP11 and our predicted three-dimensional structure of PRRSV NSP11 show that NendoU is an important structural component of PRRSV NSP11 for exerting relevant biological functions. The existence of this domain allows NSP11 to function in viral RNA synthesis. Predicting the three-dimensional structure of PRRSV NSP11 allows for a more intuitive understanding of the distribution of the NSP11 structure. Currently, the structure of PRRSV NSP11 has not been fully resolved. Further exploration of this structure will be the first step in the rational utilization and development of NSP11.

## 9. A Crystallographic NSP11 Structure

Peng et al. [25] expressed the PRRSV NSP11 protein efficiently in *Escherichia coli* and obtained crystals with a high diffraction rate through the careful screening of the crystal growth conditions. Their analyses revealed that the NSP11 monomer comprises 17 β-sheets and 3 α-helices.

Their studies revealed that serine at position 74 and phenylalanine at position 76 play crucial roles in NSP11 dimerization. These analyses further indicated that the dimerization of PRRSV NSP11 is imperative for its NendoU activity. Biochemical and structural data also support the notion that the stable dimeric structure is foundational to the NendoU function of NSP11. This study provides additional evidence supporting the idea that specific amino acid sites play a crucial role in driving NSP11 dimerization.

In another experiment, researchers analyzed the crystal structure of a PRRSV NSP11 K173A mutant (PDB code 5EYI) and discovered that the structure of NSP11 comprises conserved, compact N-terminal and C-terminal domains (Figure 6). This study provides novel insights into the NendoU protein family, which can enable a better understanding of the molecular mechanisms involved and aid in the development of antiviral drugs [23]. The analysis of NSP11’s crystal structure provides valuable insights into its mechanism of action.

The crystal structure study confirmed that NSP11 operates as a dimer. It also revealed that the amino acid sites responsible for NSP11 dimerization are strongly conserved. These results may offer a new structural foundation for the creation of antiviral medications.

## 10. Expression and Diagnosis of NSP11

The diagnostic challenge PRRSV presents is mainly due to its rapid evolution. The functional importance of PRRSV NSPs in viral replication and host immunity makes them an attractive target for diagnostic test development. Further research on NSP11 will help develop more sensitive and specific diagnostic assays.

Liu [80] used PRRSV NSP11 expressed in *E. coli* BL21(DE3) to prepare monoclonal antibodies against NSP11. This NSP11 monoclonal antibody can be used as a primary antibody for indirect immunofluorescence experiments and can also be used for PRRSV diagnosis. In their study, Contreras-Luna et al. [81] utilized recombinant proteins to generate hyperimmune sera and conducted serological assays to verify the presence of neutralizing antibodies. Their findings revealed that antibodies against PRRSV NSPs remained detectable up to 4 days after infection. Additional serological tests confirmed that the NSP11 recombinant protein was effective in inducing the production of neutralizing antibodies against PRRSV. Recombinant NSP11 has been found to enhance immunogenicity in pigs and exhibit high specificity in identifying antigens on PRRSV-positive pig farms. This high specificity and sensitivity are helpful to detect PRRSV via the NSP11 antibody. In another experiment, An NSP7 dual enzyme-linked immunosorbent assay (ELISA) was used as a differential test of PRRSV serology with high levels of sensitivity and specificity [82]. An ELISA is an additional tool for routine or follow-up diagnoses, and offers substantial value for epidemiological surveys and outbreak investigations. This suggests that PRRSV non-structural proteins can be used as detection targets and suggests that NSP11 may also be detected using a double ELISA.

The N protein is the most abundant and important structural protein in PRRSV, playing a key role in viral assembly [83]. It is a critical target for vaccine development, as it can induce both cellular and humoral immune responses [84]. In contrast, the NSP11 protein only induces humoral immune responses and has relatively weak immunogenicity. Choosing NSP11 as a target for vaccine development may lead to increased vaccine safety and stability and reduce vaccine side effects, and its use as a diagnostic target may reduce misdiagnoses. NSP11 is an endoribonuclease that can cleave RNA; thus, it has the potential to be a therapeutic drug target for PRRSV RNA. NSP11 can interact with various host cellular proteins, making it a promising immunomodulatory agent or biomarker for detecting host immune responses following a PRRSV infection.

In summary, research on NSP11 will not only help understand its important role in PRRSV replication and pathogenesis but also provide new ideas and methods for PRRSV diagnosis and identification, such as the development of a NSP11-based ELISA and monoclonal antibodies. PRRSV NSP11 has important biological functions. The current research on PRRSV NSP11 shows promising application potential as a diagnostic reagent and drug development target. However, little research has been done in this field to date. No NSP11-targeted drugs or commercial diagnostic assays have been developed. Nevertheless, recombinant NSP11 and its antibodies have demonstrated potential value for diagnostic applications.

## 11. Future Perspectives

PRRSV NSP11 possesses a unique NendoU structure, which is conserved and essential for its biological functions. Genetic and evolutionary analyses of this study confirm this viewpoint, indicating that PRRSV NSP11 has a high degree of nucleotide homology and is relatively conserved during evolution. Interactions between viral and host proteins are fundamental to viral survival. The NendoU structure enables NSP11 to participate in viral RNA synthesis and impact viral replication, interact with host proteins to exert NendoU functions, and influence the host’s innate immune response. Experiments including the transfection of NSP11-specific siRNAs and overexpression of NSP11 revealed that PRRSV NSP11 can participate in and affect viral replication. The interactions between the virus and host include interactions between PRRSV NSP11 and host proteins such as IRAK1, PCSK9, and TRIM59. Additionally, NSP11 can interfere with cellular pathways. The interactions between NSP11 and cellular proteins are fundamental to participating in and influencing the innate immune response. The interactions between NSP11 and host help PRRSV achieve immune escape. In summary, through interacting with multiple host proteins, PRRSV NSP11 can exert NendoU functions, participate in regulating the host’s innate immune response, and ultimately impact viral replication. These are interlinked processes. PRRSV NSP11 can interact with the PRRSV NSP2, NSP5, NSP6, NSP9, and NSP12 proteins to exert certain biological functions in tandem. Explorations of the effects of NSP11 on PRRSV virulence have confirmed that substituting NSP11 alone does not affect PRRSV virulence, although whether it will affect viral virulence in synergy with other factors requires further in-depth research. Crystal structure studies have confirmed that NSP11 forms dimers, and the amino acid sites responsible for NSP11 dimerization are highly conserved. The crystal structure analysis of PRRSV NSP11 provides new perspectives for designing antiviral drugs and a new the structural basis for drug development. In addition, recombinant NSP11 has enhanced immunogenicity and higher antigen recognition specificity in pigs. These results are of great significance for the development of diagnostic tools targeting NSP11.

The NendoU structure of NSP11 is crucial for normal viral functions and exhibits a high degree of conservation. From this perspective, NSP11 has significant research and clinical application value. To date, little research has focused on PRRSV NSP11, and its specific structure has not been completely resolved. Commercial vaccines and detection reagents targeting PRRSV NSP11 have not yet been developed. Further studies on the structure and functions of NSP11 will provide an important theoretical basis for elucidating the pathogenic mechanism and immune characteristics of PRRSV and provide new insights for vaccine development, clinical prevention, and control of PRRS.

## 12. Conclusions

Analyses of nucleotide homology and phylogenetic trees indicate strong conservation of the NSP11 sequence. The NendoU structure of NSP11 enables it to cleave viral RNA and participate in viral replication. Experimental studies have shown that substituting NSP11 alone does not affect PRRSV virulence. Further research is required to determine whether other PRRSV structural or non-structural proteins interact within NSP11 to jointly affect viral pathogenesis. NSP11 can interact with NSP2, NSP5, NSP6, NSP9, and NSP12 to synergistically exert PRRSV’s biological functions. PRRSV NSP11 participates in and regulates the host innate immune response by interacting with multiple host proteins. The NendoU structure of PRRSV NSP11 is the structural basis by which NSP11 exerts all biological functions. Crystal structure studies confirm that NSP11 functions as a dimer, and the amino acid sites responsible for NSP11 dimerization are highly conserved. Currently, no drugs or diagnostic reagents targeting NSP11 have been developed, although some studies have found that recombinant NSP11 enhances immunogenicity and antigen recognition specificity in pigs, indicating that NSP11 can be a potential target for drug development.

## Figures and Tables

**Figure 1 vetsci-10-00451-f001:**
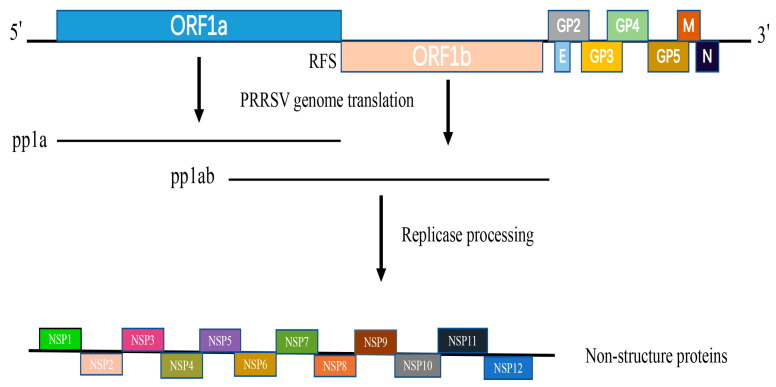
PRRSV genome structure.

**Figure 2 vetsci-10-00451-f002:**
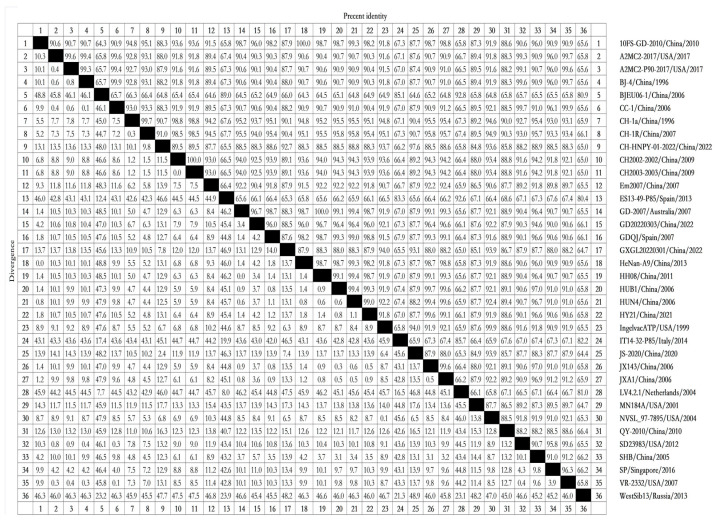
Thirty-six representative strains of PRRSV, including recent prevalent strains, were selected to obtain NSP11 nucleotide sequences for an analysis of NSP11 nucleotide homology. Clustal W in the MegAlign function of DNAStar software (version 7.0, Madison, WI, USA) was used to analyze NSP11 nucleotide sequence homology.

**Figure 3 vetsci-10-00451-f003:**
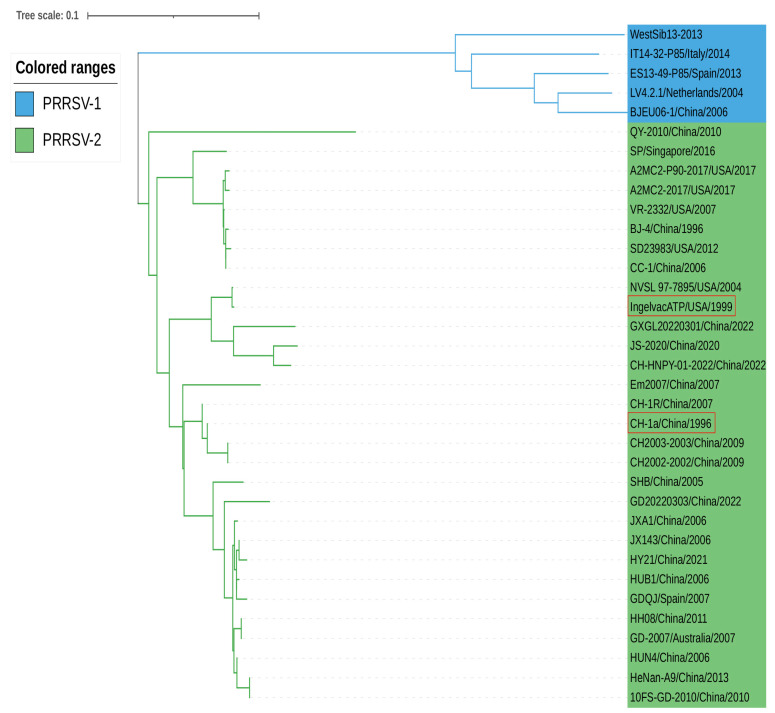
Phylogenetic analyses of the NSP11 gene were based on the sequence information of the reference strains shown in Table 1. This comparison was first performed using Clustal W in the MegAlign function of DNAStar software (version 7.0), then by using the neighbor-joining (NJ) method in MEGA software (version 7.0) with 1000 bootstrap replicates. The generated phylogenetic tree was annotated using the online software “The Interactive Tree of Life” (https://itol.embl.de, accessed on 20 June 2023). Blue represents PRRSV-1, green represents PRRSV-2, and red rectangles encircle vaccine strains.

**Figure 4 vetsci-10-00451-f004:**
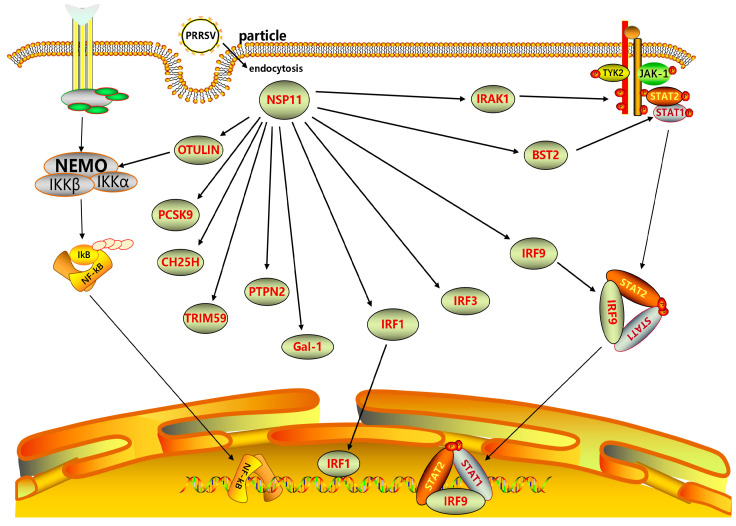
Interactions between PRRSV NSP11 and host proteins. This protein interaction network was plotted using ScienceSlides software (version 2016). Abbreviations: IRAK1, interleukin 1 receptor-associated kinase 1; IRF, interferon regulatory factor; NF-κB, nuclear factor κB; NSP, non-structural protein; PCSK9, proprotein convertase subtilisin/kexin type 9; PRRSV, porcine reproductive and respiratory syndrome virus; NSP11, non-structural protein 11; STAT2, signal transducer and activator of transcription 2; CH25H: cholesterol 25-hydroxylase; TRIM59: tripartite motif-containing 59; NEMO: NF-κB-essential modulator; IKK: inhibitor of kappa B kinase.

**Figure 5 vetsci-10-00451-f005:**
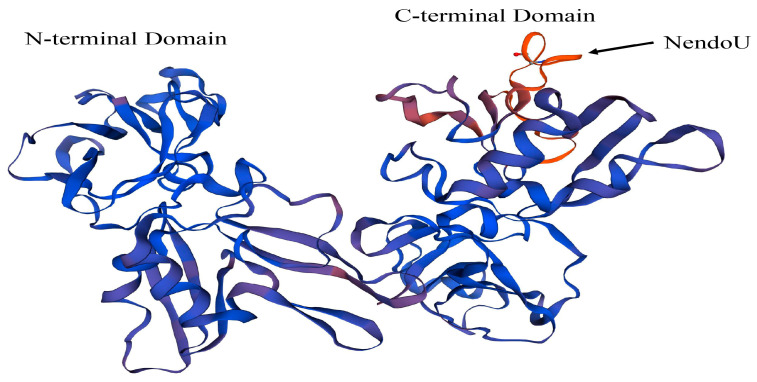
A three-dimensional structure of PRRSV NSP11 protein was predicted using the SWISS-MODEL online software. The amino acid sequence of the representative PRRSV strain CH-1a NSP11 was obtained from the NCBI database and imported into SWISS-MODEL for three-dimensional structure predictions. The reliability and quality of the predicted results were evaluated using the GMQE and QMEANDisCo Global algorithms. The score range of these algorithms is from 0 to 1, with higher values indicating higher accuracy and reliability of the predicted protein structures.

**Figure 6 vetsci-10-00451-f006:**
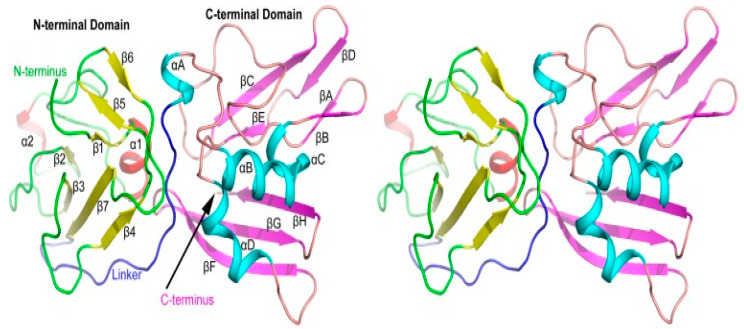
Stereoview of the PRRSV NSP11 K173A mutant structure. The whole structure consists of an NTD, a CTD, and a long linker (blue). The β-sheets and α-helices are colored yellow and red, respectively, in the NTD; and magenta and cyan, respectively, in the CTD. Modified from Zhang et al. [23].

**Table 1 vetsci-10-00451-t001:** Information about the 36 selected strains.

Year	Area	Strain	Genbank Accession Number
2022	China	GXGL20220301	OQ459664.1
2022	China	GD20220303	OQ459668.1
2022	China	CH-HNPY-01-2022	OP716076.1
2021	China	HY21	OL687155
2020	China	JS2020	MZ342900
2017	USA	A2MC2-2017	KX462792.1
2017	USA	A2MC2-P90-2017	KU318406.1
2016	Singapore	SP	AF184212.1
2014	Italy	IT14-32_P85	MK024326
2013	Spain	ES13-49_P85	MK024325
2013	China	HeNan-A9	KJ546412
2013	Russia	WestSib13	KX668221
2012	USA	SD23983	JX258843.1
2011	China	HH08	JX679179
2010	China	10FS-GD-2010	JX192634
2010	China	QY2010	JQ743666
2009	China	CH2002	EU880438
2009	China	CH2003	EU880440
2007	China	CH-1R	EU807840.1
2007	China	Em2007	EU262603.1
2007	Australia	GD-2007	EF590265.1
2007	Spain	GDQJ	GQ374441.1
2007	USA	VR2332	EF536003.1
2006	China	BJEU06-1	GU047344.1
2006	China	CC-1	EF153486.1
2006	China	HUB1	EF075945.1
2006	China	HUN4	EF635006.1
2006	China	JX143	EU708726.1
2006	China	JXA1	EF112445.1
2005	China	SHB	EU864232.1
2004	Netherlands	LV4.2.1	AY588319
2004	USA	NVSL 97-7895	AY545985.1
2001	USA	MN184A	DQ176019.1
1999	USA	Ingelvac ATP	DQ988080.1
1996	China	BJ-4	AF331831.1
1996	China	CH-1a	AY032626.1

## Data Availability

All datasets are available in the main manuscript. The dataset supporting the conclusions of this article is included within the article.

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
