# Peer review of "Research Progress on NSP11 of Porcine Reproductive and Respiratory Syndrome Virus"

_vetsci, 2023, doi:10.3390/vetsci10070451_

Round 1

Reviewer 1 Report

This review extensively analyzed the genetic evolution of NSP11, a nidovirus specific endonuclease (NendoU), effects on PRRSV replication and virulence, interaction with other PRRSV proteins and host proteins, interference of host immunity, diagnosis and the conservative characteristics of endonuclease which would provide an interesting theoretical basis for the intensive study of PRRSV pathogenesis and vaccine design in the future. This report is informative and helpful for diagnosis of PRRSV and future development of safe and potent vaccine. It is generally acceptable for publication after the following problems are solved or clarified.

1.      There are some grammatical mistakes and wordy expressions in the manuscript, which should be carefully corrected to be succinct.   

 There are some grammatical mistakes and wordy expressions in the manuscript, which should be carefully corrected to be succinct.

Author Response

Dear reviewer 1:

Thank you for constructive comments. We have made some changes to the manuscript. These changes will not influence the content and framework of the manuscript. And here we list the changes and marked in yellow in the revised manuscript. We have also responded to each of your comments and suggestions in a point-by-point manner. We appreciate for your warm work earnestly and hope that the correction will meet with approval.

1.There are some grammatical mistakes and wordy expressions in the manuscript, which should be carefully corrected to be succinct.

Response: We apologize for the grammatical mistakes and wordy expressions in the manuscript. We have asked a professional English editing company (https://www.editage.cn/) to revise the writing, please check.

Best wishes!

Xingang Yu

Reviewer 2 Report

The review report authored by Zheng et al., provided considerable exploration on NSP11 of PRRSV. Overall the review article is well written and pointed out all the mechanisms of the protein. However, the review missing few important element regarding protein genome map, structure, deep view on drug and diagnostic analysis. The necessary correction and clarification are need to improve the article.

Major comments
1. The authors showed Table 1 consist of 33 selected strains but there are 34 strains were given in the table, verify correctly.
2. Similarly, the authors presented 34 strains, but the "Ingelvac ATP" was not present in the similarity analysis Figure 1.
3. Line 94-95: A2MC2 and BJ-4 were closely related to VR332 with 99.7 percent identity than HUN4-2006 and HUB1-2006. Verify the maximum identity?
4. ES13 with GD2007 and NVSL with Porcilis DV have 65.6 lower percent identity than JS2020. Verify again the results figures 1.
5. Missing strain "Ingelvac ATP" in Figure 2.
6. Line 270: no match with the references, "Beura et al." is not correspond to the ref.62. So the ref.62 is Montaner Tarbes et al. I recommand the authors to recheck all the citated references in the main text.
7. Line 276: Ref. 30 is not Burgara-Estrella et al., it is correspond for Ref. 64. So, mention the ref.30 is Rancon-castelo et al. verify in all over the review article.
8. Line 286: What are the B-Cell conformational epitopes for NSP11? provide citation.
9. Line 338-342: Protein data bank ID for the crystal structure, and the NSP11 protein structure dimer as Figure.

Minor comments
1. Line 18: "NendoU" is not appropriate accronym, see line 22.
2. Line 38: "Arterivirus family" change to "Arteriviridae family"
3. Line 40: is ther 11 open reading frame in PRRS genome? verify 10 or 11 according to the previous citation (10.1186/s12985-017-0807-4) in the PRRSV.
4. Line 119: space between "ubiquitination" and "but"
5. Line 166: Modify "NSP11-NSP9 to NSP9-NSP11"
6. Line 179: provide chronological order of NSPs like NSP2, NSP5, NSP6, NSP9, and NSP12.

Author Response

Dear reviewer 2:

Thank you for constructive comments. We have made some changes to the manuscript. These changes will not influence the content and framework of the manuscript. And here we list the changes and marked in yellow in the revised manuscript. We have also responded to each of your comments and suggestions in a point-by-point manner. We appreciate for your warm work earnestly and hope that the correction will meet with approval.

  1. The review missing few important element regarding protein genome map, structure, deep view on drug and diagnostic analysis.

Response: We have added the protein genome map in line 88. We have added the three-dimensional structure predictions of PRRSV NSP11 in lines 381-407. We have provided additional insights into the potential role of NSP11 in drug development and diagnosis in lines 456-474.

  1. The authors showed Table 1 consist of 33 selected strains but there are 34 strains were given in the table, verify correctly.

Response: We have re-modified Table 1 in line 764, and removed the incorrect information regarding the discovery year of the Porcilis-DV-MLV strain in 2022, and added three newly discovered PRRSV strains from 2022. There are a total of 36 strains in Table 1, used for genetic evolution analysis of the NSP11 sequence.

  1. Similarly, the authors presented 34 strains, but the "Ingelvac ATP" was not present in the similarity analysis Figure 1.

Response: We have added the Ingelvac ATP strain in Figure 2 in line 119.

  1. Line 94-95: A2MC2 and BJ-4 were closely related to VR332 with 99.7 percent identity than HUN4-2006 and HUB1-2006. Verify the maximum identity?

Response: We have confirmed that A2MC2 and BJ-4 have a higher nucleotide identity with VR332 than HUN4-2006 and HUB1-2006, with a 99.7% identity. We have verified the maximum identity in lines 104-118 and found a larger nucleotide identity value.

The results of the verification are as follows: Strains with a nucleotide homology of 100.0% were all PRRSV-2 strains, such as 10FS-GD-2010/China/2010 and HeNan-A9/China/2013, as well as GD-2007/Australia/2007 and HH08/China/2011. The PRRSV-1 strains with the highest nucleotide homology were BJEU06-1/China/2006 and LV4.2.1/Netherlands/2004, with a nucleotide homology of 92.8%.

  1. ES13 with GD2007 and NVSL with Porcilis DV have 65.6 lower percent identity than JS2020. Verify again the results figure 1.

Response: We have confirmed that the nucleotide identity between ES13 and GD2007, and between NVSL and Porcilis DV, is 65.6% lower than that of JS2020. We have verified the nucleotide identity results of Figure 2 (formerly Figure 1) in lines 104-118 and found the minimum nucleotide identity value.

The results are as follows: The lowest nucleotide homology between PRRSV-1 and PRRSV-2 is 64.3%, which includes 10FS-GD-2010/China/2010 and BJEU06-1/China/2006, as well as He-Nan-A9/China/2013 and BJEU06-1/China/2006. The PRRSV-1 strains with the minimum nucleotide homology were ES13-49-P85/Spain/2013 and WestSib13/Russia/2013, with a nucleotide homology of 80.4%. The PRRSV-2 strains with the minimum nucleotide homology were CH-HNPY-01-2022/China/2022 and MN184A/USA/2001, with a nucleotide homology of 84.8%.

  1. Missing strain "Ingelvac ATP" in Figure 2.

Response: We have added the Ingelvac ATP strain to Figure 3 in line 134.

  1. Line 270: no match with the references, "Beura et al." is not correspond to the ref.62. So the ref.62 is Montaner Tarbes et al. I recommand the authors to recheck all the citated references in the main text.

Response: We have made the following modification in line 313: Beura et al. confirmed that NSP11 inhibits activation of the IFN-β promoter, and Montaner-Tarbes et al. research further supports this finding.

  1. Line 276: Ref. 30 is not Burgara-Estrella et al., it is correspond for Ref. 64. So, mention the ref.30 is Rancon-castelo et al. verify in all over the review article.

Response: We found that an incorrect reference was cited and reference 30 has been removed. We have made the following modification in line 319: Burgara-Estrella et al. [71] utilized bioinformatics prediction methods to scan amino acid sequences and identify potential T cell epitopes

We have checked all the cited references.

  1. Line 286: What are the B-Cell conformational epitopes for NSP11? provide citation.
  2. Line 18: "NendoU" is not appropriate accronym, see line 22.

Response: We have made the following modification in line 20: "nidovirus-specific endonuclease (NendoU)".

  1. Line 38: "Arterivirus family" change to "Arteriviridae family"

Response: We have made the following modification in line 41: " Arteriviridae family".

  1. Line 40: is their 11 open reading frame in PRRS genome? verify 10 or 11 according to the previous citation (10.1186/s12985-017-0807-4) in the PRRSV.

Response: After careful review and confirmation of the literature, we have made the following modification in line 43: The PRRSV genome encodes at least 10 open reading frames (ORFs).

  1. Line 119: space between "ubiquitination" and "but"

Response: We have added a space in line 160.

  1. Line 166: Modify "NSP11-NSP9 to NSP9-NSP11"

Response: We have changed NSP11-NSP9 to NSP9-NSP11 in line 212.

  1. Line 179: provide chronological order of NSPs like NSP2, NSP5, NSP6, NSP9, and NSP12.

Response: We have arranged the NSPs in chronological order as NSP2, NSP5, NSP6, NSP9, and NSP12 in line 226.

Regards!

Xingang Yu

Reviewer 3 Report

The manuscript gives a great and detailed review on PRRSV's NSP11 protein. The huge amount of information and the significant number of references are exemplary, but the manuscript is more-or-less a literature report: citing the relevant articles topic by topic, but doesn't even try to find connections among the different aspects.

To sum it up: the most critical point of mine is that the m.s. summarises the references of 8 different aspects of NSP11 from Chapter 3 to 10, without any reflection of the pieces of information popping up across these topics. For example: interactions with IFNs and other cytokins vs. virulence; importance of Ser74 and Phe76 substitutions in different aspects. Finally the reader get many pieces of information without a 'story', without any comprehensive idea to find connections among them. The title "Research progress..." suggested, that we would see the major steps of bioscience revealing the structure, the function, the interactions and even the significance of NSP11, but we must have been disappointed. You start with genetic analysis of similarity and phylogeny, but you not even refers to these results in the later part. E.g.: Any of the substitutions or genetic motives mentioned in the chapters 3-10 have been found in the 33 analysed sequences. Discuss it!

I suggest to reedit the chapters 11&12 and complement this final part with connecting and comprehensive paragraphs.

Here are some editing mistakes to correct for the improvement of the m.s.:

line 59: If you mention the spreading the virus, just mention the recent efforts and promising acts on eradication in Europe.

line 60-70: As a review, it would be informative to insert figure(s) on the structure of the polyprotein and the structure of NSP11, and you can refer to this discussing on domains and regions later on.

line 70: Is there any meaning to the role of NSP11 that EndoU is a relative of Xenopus' endonuclease? If yes, discuss it!

line 80: How were the 33 strains were selected? First of all: not strains, but sequences from GenBank were selected, correct? According to what criteria? Why do you mention the 3 named ones? Do they have any special importance in the review? In lines 94-97, similarly: why these 6 are named?

Fig 1 & 2: As there is a small piece of original work, but in a review there is no space for Materials & Methods. So you should write all the necessary M&M info in figure captations. What algorithm was used? Amino acid or nucleic acid- based similarity and phylogeny?

Increase the font in Figure 1, it's unreadable. Are there different similarity analyses in the two half of the matrix? 

In lines 100-101: "relatively conservative" - you say. Relative to what? Is there any reference article on how conservative should it be?

Figure 3 : What is the origin of the graphic? Or is it original work? I miss the NSP11 itself in the figure. Where is it? Six of the primarily interacting proteins have no further connections (PCSK9; CH25H; etc.). Is there any information on their role?

Minor editing errors and some strange language usage (like 'The positive rate of ...' instead of 'Rate of positivity' in line 55) occur. A review of a native speaker would improve the text, but it is already understandable.

Author Response

Dear reviewer 3:

Thank you for constructive comments. We have made some changes to the manuscript. These changes will not influence the content and framework of the manuscript. And here we list the changes and marked in yellow in the revised manuscript. We have also responded to each of your comments and suggestions in a point-by-point manner. We appreciate for your warm work earnestly and hope that the correction will meet with approval.

  1. For example: interactions with IFNs and other cytokins vs. virulence; importance of Ser74 and Phe76 substitutions in different aspects.

Response: After We have read all the references, we did not find any literature that can demonstrate whether the interactions of PRRSV NSP11 with IFNs and other cytokines are related to virus virulence.

We have added information about the effects of key amino acid mutations in lines 323-329.

  1. Finally the reader get many pieces of information without a 'story', without any comprehensive idea to find connections among them. The title "Research progress..." suggested, that we would see the major steps of bioscience revealing the structure, the function, the interactions and even the significance of NSP11

Response: We appreciate the reviewer’s comments. We have added a figure of the NSP11 gene in line 89. We have also predicted and analyzed a three-dimensional protein structure of NSP11 in lines 381-399.

We have added the following content for the connections between different sections. They are in lines 476-527.

  1. You start with genetic analysis of similarity and phylogeny, but you not even refers to these results in the later part. E.g.: Any of the substitutions or genetic motives mentioned in the chapters 3-10 have been found in the 33 analysed sequences. Discuss it!

Response: We have connected and discussed the genetic analysis of similarity and phylogeny in lines 477-479 of section 11 and line 512 of section 12.

  1. I suggest to reedit the chapters 11&12 and complement this final part with connecting and comprehensive paragraphs.

Response: Thanks for reviewer’s suggestion. We have re-edited sections 11 and 12 in lines 475-527, adding paragraphs that connect and synthesize the information. Please check.

  1. line 59: If you mention the spreading the virus, just mention the recent efforts and promising acts on eradication in Europe.

Response: We have added a description of relevant literature on the prevalence and prevention of European PRRSV in lines 62-70.

  1. line 60-70: As a review, it would be informative to insert figure(s) on the structure of the polyprotein and the structure of NSP11, and you can refer to this discussing on domains and regions later on.

Response: We have inserted a figure of the PRRSV gene structure in line 89, and conducted a prediction and analysis of the three-dimensional protein structure of NSP11 in lines 381-399.

  1. line 70: Is there any meaning to the role of NSP11 that EndoU is a relative of Xenopus' endonuclease? If yes, discuss it!

Response: Yes, we have discussed the relationship between NSP11 and Xenopus endonuclease in lines 82-87.

  1. line 80: How were the 33 strains were selected? First of all: not strains, but sequences from GenBank were selected, correct? According to what criteria? Why do you mention the 3 named ones? Do they have any special importance in the review? In lines 94-97, similarly: why these 6 are named?

Response: We have added the selection criteria for these 33 strains and sequences from GenBank in lines 98-101. Our criteria for selecting strains are as follows: Strains selected included strains identified in various years spanning 1996 through 2022, vaccine strains, and commonly cited representative strains.

Regarding the question of whether the selected items were not strains but sequences from GenBank, our answer is as follows: Yes, we selected the PRRSV NSP11 sequences from GenBank. We have changed the term "strains" to "sequences" and provided a detailed explanation of the sequence sources in line 98.

We have provided the following explanation for why we mentioned these strains and whether they hold any special significance in the review: In the original manuscript, we mentioned these three named sequences because we believed they represented either the highest or lowest nucleotide homology and provided information on the conservation of the PRRSV NSP11 gene. The six named sequences mentioned in lines 94-97 were selected because they were believed to have close or distant evolutionary relationships based on the results of the phylogenetic analysis.

However, to provide a more comprehensive analysis of genetic evolution, we have re-edited and analyzed the genetic evolution section in lines 104-133.

  1. Fig 1 & 2: As there is a small piece of original work, but in a review there is no space for Materials & Methods. So you should write all the necessary M&M info in figure captations. What algorithm was used? Amino acid or nucleic acid- based similarity and phylogeny?

Response: We used Clustal W in the MegAlign function of DNASTAR software (version 7.0, Madison, WI) to analyze NSP11 nucleotide sequence homology.

When constructing the phylogenetic tree, we first performed comparison using Clustal W in the MegAlign function of DNAStar software (version 7.0), followed by using the neighbor-joining (NJ) method in MEGA software (version 7.0) with 1000 bootstrap replicates. The generated phylogenetic tree was annotated using the online software “The Interactive Tree of Life” (https://itol.embl.de).

We performed similarity and phylogenetic analysis based on nucleotide sequences.

We have re-edited the figure captions for Figures 2 and 3 in lines 119 and 134.

  1. Increase the font in Figure 1, it's unreadable. Are there different similarity analyses in the two half of the matrix?

Response: We have increased the font size of Figure 2 in line 119 for better readability.

In the two halves of the matrix, the upper right part represents nucleotide homology, while the lower left part represents nucleotide divergence. We have selected the upper right part to analyze the nucleotide homology of PRRSV NSP11 in lines 104-118.

  1. In lines 100-101: "relatively conservative" - you say. Relative to what? Is there any reference article on how conservative should it be?

Response: We consider PRRSV NSP11 to be relatively conserved compared to PRRSV NSP2. We have made the following modification in line 141, citing relevant literature: "PRRSV NSP11 is relatively conserved compared to highly variable PRRSV NSP2".

  1. Figure 3: What is the origin of the graphic? Or is it original work? I miss the NSP11 itself in the figure. Where is it? Six of the primarily interacting proteins have no further connections (PCSK9; CH25H; etc.). Is there any information on their role?

Response: Figure 4 (former Figure 3) is our original work. We have edited the figure caption to include the source of the image and added NSP11 protein to the figure.

To confirm whether these six primarily interacting proteins have further connections and to obtain information on their roles, we used the STING software to predict protein-protein interactions. Our results showed that only an interaction between STAT2 and IRF9 was detected.

  1. Comments on the Quality of English Language Minor editing errors and some strange language usage (like 'The positive rate of ...' instead of 'Rate of positivity' in line 55) occur. A review of a native speaker would improve the text, but it is already understandable.

Response: We have replaced "The positive rate of" with "Rate of positivity" in line 58.

We apologize for editing errors and some strange language usage. We have asked a professional English editing company (https://www.editage.cn/) to revise the writing, please check.

Regards!

Xingang Yu

Reviewer 4 Report

Overall Comments:

The article titled " Research progress on NSP11 of porcine reproductive and respiratory syndrome virus” aimed to conduct a review to assess the crucial protein NSP11 in the biology of Porcine reproductive and respiratory syndrome virus (PRRSV) and its effects on replication, virulence, and interaction with other PRRSV proteins and host proteins. In addition, the article also discusses the conservative characteristics of enzyme activity and diagnosis. The article provides important theoretical basis for an in-depth study of PRRSV pathogenesis and vaccine design.

Overall, this review had a good structure and summarized some interesting results about PRRSV NSP11 protein. However, minor issues need to be improved before the manuscript can be recommended for publication.

11)   The legends in Figure 2 were not informative.

22)   Table 1 is not suitable for inclusion in the main text, it should be uploaded as supplementary document. In addition, each strain in the table lacks the corresponding references.

3 3) The author lacks the necessary summary after listing the relevant research results in the main text (such as Section 3, 4, 7). Please merge some paragraphs and make appropriate summaries accordingly.

Author Response

Dear reviewer 4:

Thank you for constructive comments. We have made some changes to the manuscript. These changes will not influence the content and framework of the manuscript. And here we list the changes and marked in yellow in the revised manuscript. We have also responded to each of your comments and suggestions in a point-by-point manner. We appreciate for your warm work earnestly and hope that the correction will meet with approval.

  1. The legends in Figure 2 were not informative.

Response: We have made modifications to the legend of Figure 3 in lines 135-140.

  1. Table 1 is not suitable for inclusion in the main text, it should be uploaded as supplementary document. In addition, each strain in the table lacks the corresponding references.

Response: Due to system limitations, Table 1 cannot be uploaded as a separate attachment. We have moved Table 1 to the end of the manuscript, in line 764. In addition, we have supplemented Table 1 with corresponding references for each strain in line 764.

  1. The author lacks the necessary summary after listing the relevant research results in the main text (such as Section 3, 4, 7). Please merge some paragraphs and make appropriate summaries accordingly.

Response: We have summarized Section 3 in lines 173-180 as follows: Wen et al. and others (including a study of NSP11 function using P21, transfection of NSP11-specific siRNA, and overexpression of NSP11) have confirmed that PRRSV NSP11 participates in and affects viral replication. PRRSV NSP11 is able to participate in and affect viral replication. In addition, the special NendoU structure of PRRSV NSP11 directly determines the important role NSP11 plays in normal PRSSV life activities. In-depth studies of PRRSV NSP11 are necessary for the future prevention and control of PRRS, and for research and development of related drugs and vaccines. Therefore, we speculate that inhibitors targeting NSP11 may become a powerful tool for addressing PRRSV infection.

We have summarized Section 4 in lines 203-208 as follows: Current research indicates that replacing NSP11 alone or simultaneously replacing NSP11 and NSP10 did not affect PRRSV virulence. However, the study by Li et al. mentioned above found that replacing NSP9 or NSP10 alone did not affect viral pathogenicity, but replacing NSP9 and NSP10 at the same time did affect viral virulence. We thus speculate that other factors, including structural proteins or non-structural PRRSV proteins may jointly affect viral virulence with NSP11. Extensive research is required to confirm this.

We have summarized Section 7 in lines 344-354 as follows: The research described above provides ample evidence that PRRSV NSP11 can participate in the host's innate immune response. The relationship between PRRSV NSP11 and host innate immunity is undoubtedly a focus of research with respect to PRRSV NSP11 function. Through the above research on interactions between PRRSV NSP11 and various cytokines participating in and affecting host innate immunity, we found that NSP11 can participate in various cellular pathways to influence normal transmission of biological information in cells. For example, interacting with IRF3 helps PRRSV escape the host's innate immune system, inhibiting ERK signaling, resulting in TNF-α suppression, impacting IL-17, and indirectly impacting the host's natural immune system. The immune escape of PRRSV is an important reason why PRRS is difficult to prevent and control. NSP11 as a drug design target may be a solution to PRRSV immune evasion.

We have summarized Section 8 in lines 400-407 as follows: The above research on the NendoU structure within PRRSV NSP11 and our predicted three-dimensional structure of PRRSV NSP11 support that NendoU is an important structural component of PRRSV NSP11 for exerting relevant biological functions. The existence of this domain allows NSP11 to function in viral RNA synthesis. Predicting the three-dimensional structure of PRRSV NSP11 allows for a more intuitive understanding of the distribution of the NSP11 structure. Currently, the structure of PRRSV NSP11 has not been fully resolved. Further exploration of this structure will be the first step in the rational utilization and development of NSP11.

We have summarized Section 10 in lines 456-474 as follows: The N protein is the most abundant and important structural protein in PRRSV, playing a key role in virus assembly [83]. It is a critical target for vaccine development, as it can induce both cellular and humoral immune responses [84]. In contrast, the NSP11 protein only induces humoral immune responses and has relatively weak immunogenicity. Choosing NSP11 as a target for vaccine development may increase vaccine safety and stability, reduce vaccine side effects, and its use as a diagnostic target may reduce misdiagnoses. NSP11 is an endoribonuclease that can cleave RNA, and thus has the potential to be a therapeutic drug target for PRRSV RNA. NSP11 can interact with various host cellular proteins, making it a promising immunomodulatory agent or biomarker for detecting host immune responses following a PRRSV infection.

In summary, research on NSP11 will not only help understand its important role in PRRSV replication and pathogenesis but also provide new ideas and methods for PRRSV diagnosis and identification, such as the development of an NSP11-based ELISA and monoclonal antibodies. PRRSV NSP11 has important biological functions. Current research on PRRSV NSP11 shows promising application potential as a diagnostic reagent and drug development target. However, little research has been done in this field to date. No NSP11-targeted drugs or commercial diagnostic assays have been developed. Nevertheless, recombinant NSP11 and its antibodies have demonstrated potential value for diagnostic applications.

We have merged some paragraphs in Sections 3, 6, and 7.

Regards!

Xingang Yu

Round 2

Reviewer 2 Report

Thanks for all the modifications included in the revised manuscript. The authors given responses for the all the comments, and modified accordingly in the article.

Author Response

Dear reviewer 2:

I am writing to express my sincere gratitude for your time and effort in reviewing our manuscript and providing valuable feedback. We truly appreciate your recognition of the modifications we made in response to your comments and are delighted to know that you found our revisions satisfactory. We are grateful for the opportunity you have given us to improve our work.

Once again, thank you for your dedication and expertise in reviewing our manuscript. We look forward to the possibility of collaborating with you in the future.

Regards!

Xingang Yu

Reviewer 3 Report

The manuscript has been developed a lot. 

The Table 1 is referred, but it has not been inserted in the manuscript - by an editing mistake, probably.

A thorough, profound editing by a native speaker would improve the quality of the manuscript.

Author Response

Dear reviewer 3:

Thank you very much for your helpful feedback on our manuscript. We appreciate your comments and suggestions, which have been instrumental in improving the quality of our work.

We have inserted Table 1 in line 106, which was previously referred to but not inserted due to an editing mistake.

Furthermore, we have invited a native speaker to edit the manuscript. As a result, the quality of the manuscript's English has been improved. Please check.

Thank you again for your time and effort in reviewing our manuscript. We hope that the revised manuscript meets your expectations.

Regards!

Xingang Yu